# Mobile apps for detecting falsified and substandard drugs: A systematic review

Agustín Ciapponi[1,2]⊛*, Manuel Donato[1,2]⊛, A. Metin Gülmezoglu[3]⊛, Tomás Alconada[1,2]⊛, Ariel Bardach[1,2]⊛

1 Instituto de Efectividad Clínica y Sanitaria (IECS-CONICET), Buenos Aires, Argentina, 2 Centro de Investigaciones Epidemiológicas y Salud Pública (CIESP-IECS), CONICET, Buenos Aires, Argentina, 3 Department of Reproductive Health and Research, UNDP/UNFPA/UNICEF/WHO/World Bank Special Programme of Research, Development and Research Training in Human Reproduction (HRP), World Health Organization, Geneva, Switzerland

⊛ These authors contributed equally to this work.
* aciapponi@iecs.org.ar

## Abstract

The use of substandard and counterfeit medicines (SCM) leads to significant health and economic consequences, like treatment failure, rise of antimicrobial resistance, extra expenditures of individuals or households and serious adverse drug reactions including death. Our objective was to systematically search, identify and compare relevant available mobile applications (apps) for smartphones and tablets, which use could potentially affect clinical and public health outcomes. We carried out a systematic review of the literature in January 2020, including major medical databases, and app stores. We used the validated Mobile App Rating Scale (MARS) to assess the quality of apps, (1 worst score, 3 acceptable score, and 5 best score). We planned to evaluate the accuracy of the mobile apps to detect SCM. We retrieved 335 references through medical databases and 42 from Apple, Google stores and Google Scholar. We finally included two studies of the medical database, 25 apps (eight from the App Store, eight from Google Play, eight from both stores, and one from Google Scholar), and 16 websites. We only found one report on the accuracy of a mobile apps detecting SCMs. Most apps use the imprint, color or shape for pill identification, and only a few offer pill detection through photographs or bar code. The MARS mean score for the apps was 3.17 (acceptable), with a maximum of 4.9 and a minimum of 1.1. The 'functionality' dimension resulted in the highest mean score (3.4), while the 'engagement' and 'information' dimensions showed the lowest one (3.0). In conclusion, we found a remarkable evidence gap about the accuracy of mobile apps in detecting SCMs. However, mobile apps could potentially be useful to screen for SCM by assessing the physical characteristics of pills, although this should still be assessed in properly designed research studies.

## Introduction

The World Health Organization (WHO) defines substandard or "out of specification" medicines as authorized medical products that fail to meet either their quality standards or

**Funding:** This work was funded by an independent grant from HRP (the UNDP/UNFPA/UNICEF/WHO/

World Bank Special Programme of Research, Development and Research Training in Human Reproduction https://wwww.who.int/ reproductivehealth/hrp/en/). The funder's role was limited to developing the terms of references and reviewing the drafts. The funders had no role in the final content or decision to publish the manuscript.

**Competing interests:** The authors have declared that no competing interests exist.

specifications, or both [1]. Counterfeit or falsified medicines are products that deliberately or fraudulently misrepresent their identity, composition or source. Poor-quality medicines have important adverse health consequences, including the potential for treatment failure, the development of antimicrobial resistance, and serious adverse drug reactions, including death [2]. Apart from safety and effectiveness concerns, substandard and counterfeit medicines (SCMs) can carry economic costs such as the treatment of adverse events by the health system and resources wasted, leading to complications that are borne by consumers, facilities and third-payers. Yet, there are also potential indirect costs to consider, such as loss of productivity due to extra-days of illness, and reduced sales and tax revenues coming from regular medicines [3,4]. SCMs not only leads to public health and economic consequences, but it also weakens efforts, for example, to attain the United Nations Sustainable Development Goal related to achieving universal access to safe and effective care, including essential medicines [5].

Quality assurance of medicines represents a significant challenge for governments, regulators, and pharmaceutical companies at a global level [6,7]. That is why the only way to fight against this problem is a multifaceted approach with all the actors involved participating and targeting various levels of the pharmaceutical supply chain from developers to consumers [8–10]. With this purpose, a variety of technologies from mobile apps and handheld devices to sophisticated analytical chemistry methods, have been developed to detect SCMs [11]. Mobile health (mHealth) is a general term for the use of mobile phones and other wireless technology in medical care. Some software applications (apps) may be able to identify authorized medicines and discriminate against SCMs by detecting differences in several aspects, such as shape, color and others. These apps generally have a database of visual characteristics of currently authorized medicines and use a phone camera to compare it against the sample product to evaluate. Also, some technologies, mostly desktop applications, rely on the same database to detect drug inconsistencies [12,13].

The full extent of the SCM problem is largely unknown, and scientific research is variable and of poor methodological quality [14]. However, an increase in SCM is a growing global concern. If an inexpensive and widely available technology such as an app can help in the screening and possible detection of SCM, it could have significant potential use. Our objective was to systematically search, identify, synthesize and compare relevant mobile apps whose use could beneficially impact on public health decision-making.

## Methods

We conducted a systematic review of the literature following the Cochrane methods and the PRISMA guidelines for reporting in S1 File [15,16]. Literature searches designed by a trained librarian were conducted in January 2020 in the Cochrane Database of Systematic Reviews (CDSR), Cochrane Central Register of Controlled Trials (CENTRAL), Database of Abstracts of Reviews of Effects (DARE), MEDLINE, EMBASE, Clinicaltrials.gov, the WHO International Clinical Trials Registry Platform (ICTRP), MedRxiv and SciFinder. The basic search strategy designed for Medline (PubMed) included the following terms: (Mobile Applications [Mesh] OR Mobile App*[tiab] OR Electronic App*[tiab] OR Portable App*[tiab] OR Software App*[tiab] OR Mobile Software[tiab] OR Portable Software[tiab] OR Mobile Based[tiab] OR Medical App*[tiab] OR Mobile Authentic*[tiab] OR Cell Phone[Mesh] OR Mobile Phone* [tiab] OR Cell Phone*[tiab] OR Mobile Telephon*[tiab] OR Cell Telephon*[tiab] OR Cellular Phone*[tiab] OR Cellular Telephon*[tiab] OR Smartphone*[tiab] OR QR[tiab] OR Computers, Handheld[Mesh] OR Handheld Device[tiab] OR iPad[tiab] OR Pill Identificat*[tiab]) AND (Counterfeit Drugs[Mesh] OR Counterfeit[tiab] OR Counterfeit[tiab] OR Fake[tiab] OR Adulterated[tiab] OR Imitation*[tiab] OR Fraudulent[tiab] OR spurious [tiab]). The search

terms were modified to suit the requirements of particular databases as was detailed in S2 File. We also searched non-peer reviewed technical reports and other online information, including Apple's App Store, Google Play Store, and a directed search in Google Scholar. We searched references of included articles and relevant literature reviews.

The strategy for Google Scholar included the following terms: *Pill Identifier Tool Medication OR Medicine OR Drug OR Falsified OR Counterfeit OR Substandard OR Surveillance OR Authentication OR Mobile OR Software.*

The systematic review protocol was registered in the PROSPERO (CRD42020163075) database in January 2020. Regarding primary eligibility criteria, we included apps claiming to analyze authorized medicines or SCMs identification in their description, aimed to directly detect and recognize solid oral medications, published in English, Spanish or Portuguese language, and available to download from the mentioned app stores. Mobile apps were excluded if they targeted non-human medications, identified illicit drugs, or examined herbals or Chinese medicines. After screening the results for each search database, the selected app names were added. If an app was listed in both stores we report them with a single value after consensus. We downloaded and installed the remaining apps as the first step to assess their eligibility for this review. Apps failing to launch in the test devices were excluded. All Apple test devices ran iPhone operating system (iOS, Apple Inc) 13.3, and all Android test devices ran Android 10.0. We also extracted general information and relevant secondary features that the apps offer, such as information provided about authorized medicines, security and privacy-related features, data sharing and social media and technical support.

All unique articles were independently assessed by two reviewers (MD and TA) based on title and abstract. Those marked for inclusion, or whose title and abstract were not sufficient to determine inclusion, were then reviewed using the full text. Data extraction and risk of bias (quality) assessment were also performed independently by these reviewers, with oversight from two senior reviewers (AC and AB). The risk of bias was planned to be assess by the Cochrane tool for Randomized Controlled Trials (RCTs) designs [17]. For other designs, including cross-sectional or cohorts, we used the NIH Study Quality Assessment Tools [18]. For content analysis, we used the Mobile App Rating Scale (MARS) that is a simple, objective, and reliable tool for classifying and assessing the quality of mobile apps [12]. The MARS app quality is a 19-item, expert-based rating scale to assess the quality of mobile apps. Each question from MARS uses a 5-point scale (1 = inadequate, 2 = poor, 3 = acceptable, 4 = good, and 5 = excellent). This expert scale consists of multiple dimensions that assess different quality aspects of apps, including end-user engagement features, aesthetics and content quality. Comprehensiveness and accuracy of the content and information of the app for the MARS are assessed on questions #15 and #16. We completed all phases of the study selection using COVIDENCE®, a web-based platform designed for the processing of systematic reviews [19]. Authors of articles were contacted to obtain missing or supplementary information when necessary.

A pre-designed general data extraction form was used after pilot testing. We resolved disagreements during all phases by consensus by the two initial reviewers (MD and TA) and, when necessary, a third reviewer (AC or AB) decided on them if a consensus was not reached. We extracted the following: general information about the study (publication type, year of publication, journal, authors' names, and language), research location (geographical region, country, province, city, and setting) and study population (sample size, age at enrollment, living in rural or urban area under, and dates of initiation and ending of data collection). We performed descriptive analyses of the extracted data, and we structured data in tables, to describe the mobile apps for detecting SCMs which included a set of both general information and relevant secondary features of the apps, and which were available in the app stores.

## Results

Our search strategy retrieved 335 references through the literature search in databases and 42 reports of technologies coming from other sources, such as Apple's App Store, Google Play Store and Google search engines. We finally included 25 mobile apps, 16 websites that compare the given medication with large internal databases, one preprint study evaluating one mobile app and other study comparing four mobile apps (Fig 1), along with a detail of their main characteristics to facilitate comparisons. These reports referred to the usability and the ability of apps to correctly identify pills. However, did not find direct assessment of SF products in the real-world.

Considering the nature of the data, we could not perform a meta-analysis.

Table 1 and Fig 2 summarize the general characteristics of the reviewed apps and the app quality mean MARS scores available on platforms. Of the 25 apps, eight were from Apple's App Store, eight from Google Play, eight were listed in both stores and one (MedSnap) on none of the platforms, but was identified through Google Scholar and MedRxiv. Twenty-two developers developed the 24 apps available on platforms, and 17 were currently available to detect pills without any cost. Seven were available after purchase or offered more options through a subscription. The last update dates for the Android apps were from February 2010 to March 2020, whereas the iOS apps were generally the most frequently updated, and dates ranged from October 2015 to March 2020. The average rating for the apps was between acceptable and good, with a mean score of 3.78, ranging from a minimum of 1 to a maximum of 5, but sometimes it was not available.

The size of the circles is proportional to the quintiles of MARS mean score, and their color identifies the platform source. The axes of the graph indicate the average user ratings and the number of raters.

Most apps claim to detect pills by evaluating the imprint, color or shape, and only a few offer the possibility of identification through a photograph or bar code. The app quality mean

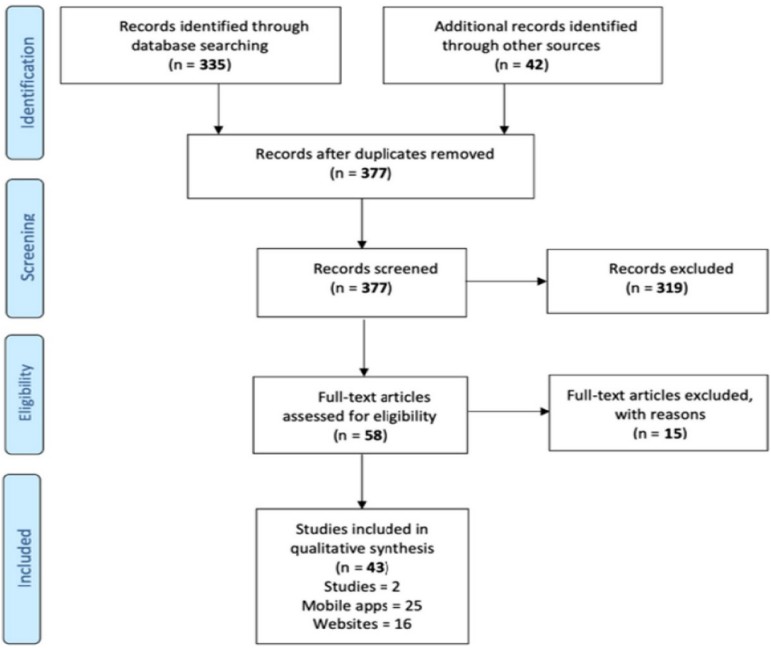

**Fig 1. Records flow diagram.**

**Table 1.  Characteristics of included mobile apps for detecting SCMs.**

| Name | Platform | Developer | Version | Last update | Cost | Average user Rating (out of 5) | Engagement (out of 5) | Functionality (out of 5) | Aesthetics (out of 5) | Information (out of 5) | MARS mean score (out of 5) |
|---|---|---|---|---|---|---|---|---|---|---|---|
| Advanced Pill Identifier & Drug Info | Android | appmaniateam | 1.6 | 21/8/18 | Free | 3.7 | 3.4 | 4.3 | 3.3 | 3.5 | 3.6 |
| CheckFake | Android | dlt.sg Apps | 1.0 | 14/4/18 | Free | 4.4 | 1.8 | 1 | 1.3 | 1 | 1.3 |
| Drug Facts by PillSync.com | iOS | Scanidme inc | 5.2 | 26/6/19 | Free | 4.1 | 3.4 | 3.2 | 3 | 3.3 | 3.2 |
| Drug Facts Pill ID | iOS | Scanidme inc | 5.2 | 1/7/19 | Free | NA | 3.4 | 3.2 | 3 | 3.3 | 3.2 |
| Drug Interaction Checker | iOS/ Android | HYDL | 13 | 1/2/19 | Free | NA | 3 | 3 | 3 | 3 | 3 |
| Drug Search App | Android | Drug Search App | 1.6 | 14/9/18 | Free | 3.6 | 1.4 | 1 | 1 | 1 | 1.1 |
| Drugs.com Medication Guide | iOS/ Android | Drugs.com | 2.9.7 | 31/10/19 | Free | 4.9 | 4.2 | 4.5 | 4.3 | 4.3 | 4.3 |
| Epocrates | iOS/ Android | Epocrates, Inc. | 20.1 | 13/2/20 | Free | 4.3 | 4.8 | 4.8 | 5 | 4.4 | 4.7 |
| IBM Micromedex Drug Reference | iOS/ Android | Micromedex | 2.1b815 | 19/3/20 | 2.99 $ annual | 4.3 | 5 | 5 | 5 | 4.6 | 4.9 |
| iNarc: Pill Finder and Identifier | iOS | Amit Barman | 4 | 1/10/15 | 0.99 $ | NA | 1.6 | 2 | 2.3 | 2 | 2 |
| Lexicomp | iOS/ Android | Lexicomp | 5.3.2 | 18/3/20 | 799 $ annual | 4.1 | 5 | 5 | 5 | 5 | 5 |
| Medscape | iOS/ Android | WebMD, LLC | 7.3.1 | 10/2/20 | Free | 4.6 | 5 | 5 | 5 | 4.6 | 4.9 |
| MedSnap | Own platform | MedSnap | NA | NA | Not free | NA | 4 | 4.2 | 3.7 | 4.1 | 4 |
| Pepid | iOS/ Android | Pepid, LLC | 6.2 | 20/3/20 | Free | NA | 2 | 2.5 | 2 | 2.3 | 2.2 |
| Pill identifier | Android | Giant Brains Software | 2.6 | 28/8/18 | 0.99 $ annual | NA | 3.6 | 4 | 3.7 | 3.5 | 3.7 |
| Pill identifier | Android | Walhalla Dynamics | 7.1.1662. r | 1/2/10 | Free | 3.1 | 3.6 | 4 | 3.7 | 3.5 | 3.7 |
| Pill Identifier and Drug List | iOS | Mobixed LLC | 3.9 | 17/9/19 | Free | NA | 3 | 3.5 | 4 | 3.3 | 3.4 |
| Pill Identifier by Drugs.com | iOS | Drugs.com | 2.97 | 12/2/20 | 0.99 $ | 1 | 3.2 | 4.2 | 3.3 | 3.5 | 3.5 |
| Pill Identifier Mobile App | iOS | Eric Phung | 2.5 | 1/3/19 | 1.99 $ | NA | 1.6 | 1.7 | 1.7 | 1.5 | 1.6 |
| Pill Identifier Pro and Drug Info | Android | Mobilicks | 1.0.3 | 15/2/19 | Free | 3.7 | 3.2 | 4 | 3.7 | 3.2 | 3.5 |
| pill+: Prescription Pill Finder and Identifier | iOS | Amit Barman | 4 | 2/10/15 | 0.99 $ | 5 | 1.6 | 2 | 2 | 1.5 | 1.8 |
| PillFinder 2.0 | iOS | MedApp sp. Zo.o. | 2.0.1 | 1/3/16 | Free | NA | 2 | 2.2 | 2.3 | 2.2 | 2.1 |

*(Continued)*

**Table 1.** (*Continued*)

| Name | Platform | Developer | Version | Last update | Cost | Average user Rating (out of 5) | Engagement (out of 5) | Functionality (out of 5) | Aesthetics (out of 5) | Information (out of 5) | MARS mean score (out of 5) |
|------|----------|-----------|---------|-------------|------|--------------------------------|------------------------|--------------------------|------------------------|------------------------|----------------------------|
| Pillid.com | iOS/ Android | Douglas McKalip | 1.2.1 | 7/9/17 | Free | 3.8 | 2.6 | 4 | 2.3 | 2.7 | 2.9 |
| Prescription Pill Identifier | Android | Giant Brains Software | 2.5 | 10/9/18 | Free | 3.8 | 2.6 | 3.5 | 2.6 | 3.3 | 3 |
| Smart Pill Identifier | Android | iConiq Studios | 0.1.2 | 5/4/19 | Free | 2.2 | 1.6 | 3.5 | 3 | 2.1 | 2.6 |

MARS score (Table 1) of the 25 apps was good with 3.17, with a maximum of 4.9 (excellent) for Medscape and IBM Micromedex Drug Reference, and a minimum of 1.1 (inadequate) for Drug Search App. The mean scores of the four dimensions of MARS were examined to investigate the magnitude of the differences in quality in each dimension (Table 1). The functionality dimension resulted in the highest mean score (3.4), whereas the engagement dimension and the information dimension showed the lowest average score (3.0). It should be noted that of the 25 apps, only 4 were verified by evidence in published scientific literature, these being MedSnap, Epocrates, Medscape and IBM Micromedex (Table 2) [20,21]. On the other hand, all of the websites (16) identified out of these platforms are intended for the general population (Table 3). Nevertheless, all the websites and available software use imprint, color or shape to identify the pills.

Relevant secondary features for each app available on platforms were extracted and shown in Fig 3. The overall number of apps with drug database access was 12 (out of 25), and 9 were free to access. Drug history tracking was available in only 6 apps where one has some cost, 16 apps allow sharing while 5 have some cost, and 7 apps require login and provide password protection with only 2 having any cost.

Our search found a cross-sectional study that intended to compare Epocrates, Medscape, IBM Micromedex and Google for clinical management information designed for healthcare

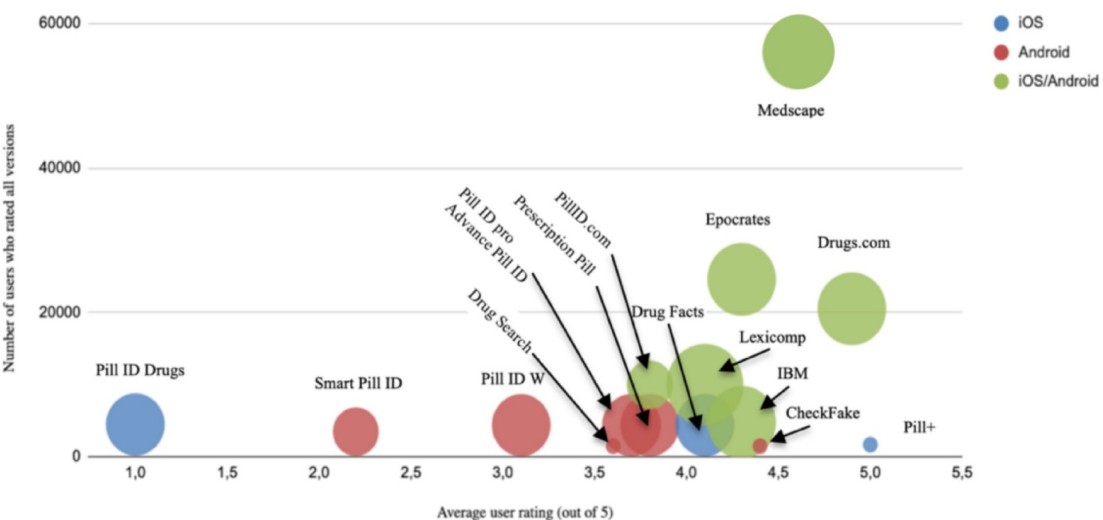

**Fig 2. Mean MARS score per mobile app by users' ratings.**

**Table 2. Detailed MARS score for all included apps.**

| App Name | Platform | 1. Entertainment | 2. Interest | 3. Customization | 4. Interactivity | 5. Target Group | 6. Performance | 7. Ease of Use | 8. Navigation | 9. Gestural Design | 10. Layout | 11. Graphics | 12. Visual Appeal | 13. Accuracy | 14. Goals* | 15. Accuracy | 16. Comprehensiveness | 17. Visual* | 18. Credibility | 19. Evidence Base |
|---|---|---|---|---|---|---|---|---|---|---|---|---|---|---|---|---|---|---|---|---|
| Advanced Pill Identifier & Drug Info | Android | 3 | 4 | 3 | 3 | 4 | 4 | 5 | 4 | 4 | 4 | 3 | 3 | 4 | 3 | 3 | 3 | 4 | 4 | NA |
| CheckFake | Android | 1 | 1 | 2 | 2 | 3 | 1 | 1 | 1 | 1 | 1 | 1 | 1 | 1 | 1 | 1 | 1 | 1 | 1 | NA |
| Drug Facts by PillSync.com | iOS | 4 | 4 | 3 | 3 | 3 | 4 | 3 | 3 | 3 | 3 | 3 | 3 | 4 | 3 | 3 | 3 | 3 | 4 | NA |
| Drug Facts Pill ID | iOS | 4 | 4 | 3 | 3 | 3 | 4 | 3 | 3 | 3 | 3 | 3 | 3 | 4 | 3 | 3 | 3 | 3 | 4 | NA |
| Drug Interaction Checker | iOS/Android | 3 | 3 | 3 | 3 | 3 | 3 | 3 | 3 | 3 | 3 | 3 | 3 | 3 | NA | NA | NA | NA | 3 | NA |
| Drug Search App | Android | 1 | 1 | 1 | 1 | 3 | 1 | 1 | 1 | 1 | 1 | 1 | 1 | 1 | 1 | NA | NA | NA | 1 | NA |
| Drugs.com Medication Guide | iOS/Android | 4 | 4 | 4 | 4 | 5 | 5 | 4 | 5 | 4 | 5 | 4 | 4 | 5 | 4 | 4 | 4 | 5 | 4 | NA |
| Epocrates | iOS/Android | 5 | 5 | 5 | 4 | 5 | 5 | 4 | 5 | 5 | 5 | 5 | 5 | 5 | 5 | 5 | 5 | 5 | 4 | 2 |
| IBM Micromedex Drug Reference | iOS/Android | 5 | 5 | 5 | 5 | 5 | 5 | 5 | 5 | 5 | 5 | 5 | 5 | 5 | 5 | 5 | 5 | 5 | 5 | 2 |
| iNarc: Pill Finder and Identifier | iOS | 1 | 1 | 2 | 2 | 2 | 2 | 2 | 2 | 2 | 2 | 2 | 3 | 2 | NA | NA | 2 | 2 | 2 | NA |
| Lexicomp | iOS/Android | 5 | 5 | 5 | 5 | 5 | 5 | 5 | 5 | 5 | 5 | 5 | 5 | 5 | 5 | 5 | 5 | 5 | 5 | NA |
| Medscape | iOS/Android | 5 | 5 | 5 | 5 | 5 | 5 | 5 | 5 | 5 | 5 | 5 | 5 | 5 | 5 | 5 | 5 | 5 | 5 | 2 |
| MedSnap | Own platform | 4 | 4 | 4 | 4 | 4 | 5 | 4 | 4 | 4 | 4 | 3 | 4 | 5 | 4 | 4 | 4 | 5 | 3 | 3 |
| Pepid | iOS/Android | 2 | 2 | 2 | 2 | 2 | 3 | 2 | 2 | 2 | 2 | 2 | 2 | 2 | 2 | 2 | 3 | 3 | 2 | NA |
| Pill Identifier and Drug List | iOS | 3 | 3 | 3 | 3 | 3 | 3 | 4 | 3 | 4 | 4 | 4 | 4 | 4 | 3 | 3 | 4 | 3 | 3 | NA |
| Pill Identifier by Drugs.com | iOS | 3 | 3 | 3 | 3 | 4 | 4 | 5 | 4 | 4 | 4 | 3 | 3 | 4 | 2 | 3 | 4 | 4 | 4 | NA |
| Pill identifier by Giant Brains Software | Android | 4 | 4 | 3 | 3 | 4 | 4 | 4 | 4 | 4 | 4 | 4 | 3 | 3 | 4 | 3 | 3 | 4 | 4 | NA |
| Pill identifier by Walhalla Dynamics | Android | 4 | 4 | 3 | 3 | 4 | 4 | 4 | 4 | 4 | 4 | 4 | 3 | 3 | 4 | 3 | 3 | 4 | 4 | NA |
| Pill Identifier Mobile App | iOS | 1 | 1 | 2 | 2 | 2 | 2 | 1 | 2 | 2 | 1 | 2 | 2 | 2 | NA | NA | NA | NA | 1 | NA |
| Pill Identifier Pro and Drug Info | Android | 3 | 3 | 3 | 3 | 4 | 4 | 4 | 4 | 4 | 4 | 4 | 3 | 3 | 3 | 3 | 3 | 3 | 3 | NA |
| pill+: Prescription Pill Finder and Identifier | iOS | 1 | 1 | 2 | 2 | 2 | 2 | 2 | 2 | 2 | 1 | 2 | 2 | 2 | NA | 1 | 1 | 2 | 1 | NA |
| PillFinder 2.0 | iOS | 2 | 3 | 1 | 2 | 2 | 2 | 2 | 2 | 3 | 2 | 3 | 2 | 1 | NA | NA | 2 | 2 | 4 | NA |
| Pillid.com | iOS/Android | 3 | 3 | 2 | 2 | 3 | 4 | 4 | 4 | 4 | 3 | 2 | 2 | 3 | 3 | 2 | 2 | 3 | 3 | NA |
| Prescription Pill Identifier | Android | 3 | 4 | 1 | 1 | 4 | 3 | 4 | 3 | 3 | 3 | 3 | 3 | 4 | 3 | 3 | 3 | 4 | 3 | NA |
| Smart Pill Identifier | Android | 2 | 2 | 1 | 1 | 2 | 2 | 4 | 4 | 4 | 3 | 3 | 3 | 2 | 2 | 2 | 2 | 3 | 2 | NA |

**Table 3. Additional resources identified from the search strategy.**

| Name | Features | Function | Recipient |
|------|----------|----------|-----------|
| AARP | Webpage | Pill identifier. Search by Imprint, Shape or Color | Consumers |
| CVS Pharmacy | Webpage | Pill identifier. Search by Imprint, Shape or Color | Consumers |
| Drugs.com | Webpage | Pill identifier. Search by Imprint, Shape or Color | Consumers |
| Epocrates | Webpage | Pill identifier. Search by Imprint, Shape, Color, Scoring, Clarity, Coating or Flavor | Consumers |
| IBM Micromedex Solutions | Webpage | Pill identifier. Search by Imprint, Shape, Color or Form | Consumers |
| Medscape | Webpage | Pill identifier. Search by Imprint, Shape, Color, Scoring or Form | Consumers |
| MedSnap | Mobile app | Pill scanner | Consumers/ Regulator |
| Pepid | Webpage | Pill identifier. Search by Imprint, Shape or Color | Consumers |
| PillBox | Webpage | Pill identifier. Search by Imprint, Shape, Color, Size or DEA schedule | Consumers |
| RedCrossDrugstore | Webpage | Pill identifier. Search by Imprint, Shape, Color, Scoring, Clarity, Coating or Flavor | Consumers |
| RxID.ca | Webpage | Pill identifier. Search by Imprint, Shape, Color, Scoring, Coating, Surface or Logo | Consumers |
| RxList.com | Webpage | Pill identifier. Search by Imprint, Shape or Color | Consumers |
| RxResouse.org | Webpage | Pill identifier. Search by Imprint, Shape, Color or Scoring | Consumers |
| RxSaver | Webpage | Pill identifier. Search by Imprint, Shape, Color, Scoring or Size | Consumers |
| WebMD.com | Webpage | Pill identifier. Search by Imprint, Shape or Color | Consumers |
| WebPoisonControl | Webpage | Pill identifier. Search by Imprint, Shape or Color | Consumers |
| WellRx | Webpage | Pill identifier. Search by Imprint, Shape or Color | Consumers |

professionals for a one-week period [20]. Medical students at an academic hospital in the United States used a score for satisfaction of search and user interface based on a 1 to 5 scale, with 1 representing lowest quality and 5 describing the highest quality. The study concluded that Medscape (Satisfaction 4.92) was the most preferred free mobile app evaluated due to its interactive and educational features, followed by IBM Micromedex and Epocrates (Satisfaction 4.58 and 4.42, respectively). This study was used to complete MARS question #19 (evidence base) about whether the app had been trialed. Further, we found a preprint study whit the objective to evaluate the sensibility and specificity of MedSnap, that is not available on the app stores. In the study [21], MedSnap models were created from trusted and authentic medications and tested against samples of authentic/trusted or falsified artesunate, artemether-lumefantrine, azithromycin, and ciprofloxacin. Results were 100% sensitivity and specificity to detect authentic and counterfeit drugs from 48 samples tested. This study allowed us to complete MARS question #19 and others parts of the questionnaire. It should be noted that this app is the only one where the accuracy to detect counterfeit drugs was formally evaluated through a study.

We used the NIH Study Quality Assessment Tools for the two included studies and in both cases a poor score was obtained, which means a high risk of bias (S3 File).

## Discussion

We found 25 apps and 16 websites for pill identification potentially useful for detecting SCM. Despite the variety of available apps, there was only two scientific publications of observational studies considered as high risk of bias, that included three apps and one of them as a preprint

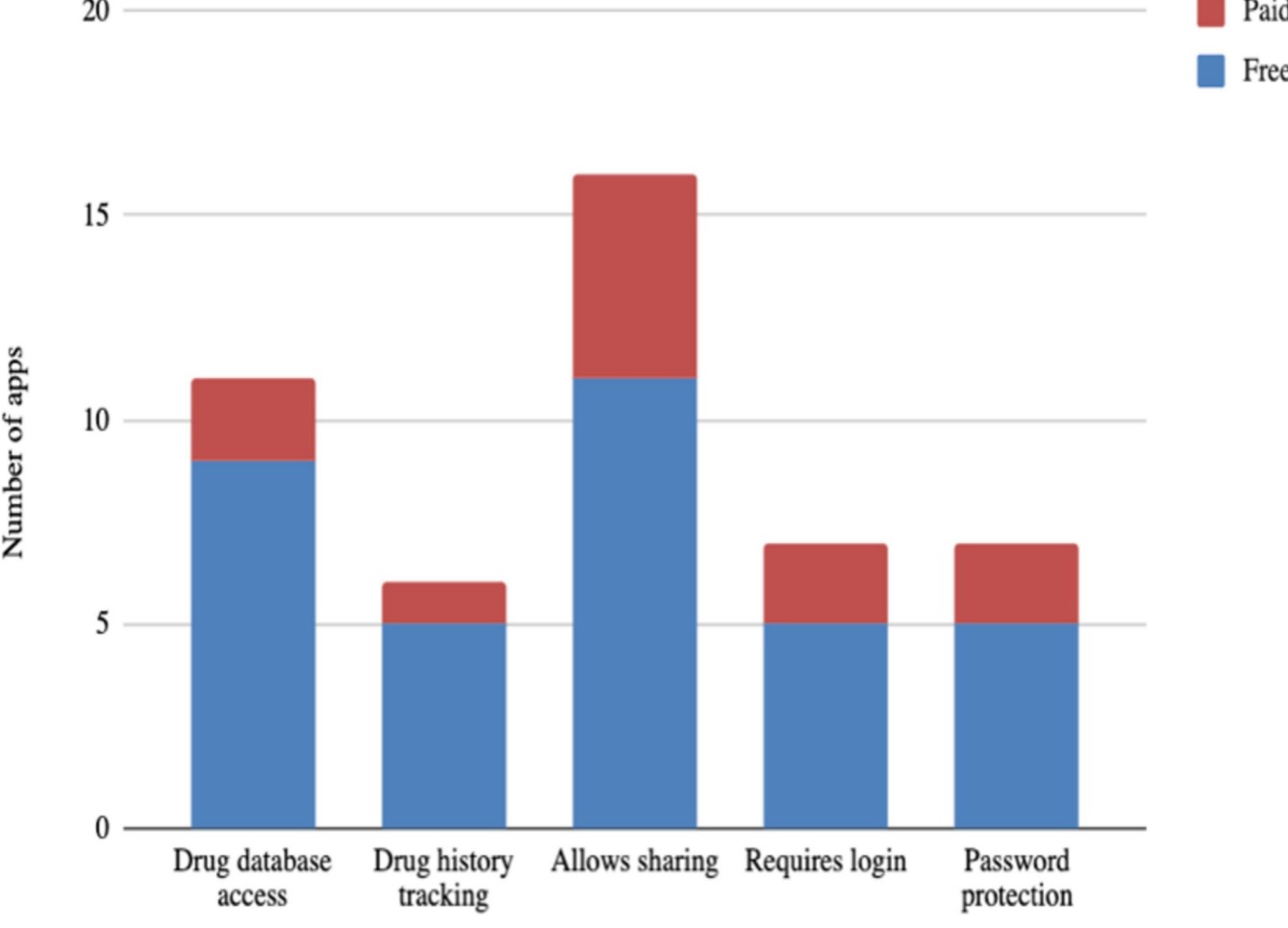

**Fig 3. Frequency of apps, available on platforms, by selective secondary features.**

report. This highlights the lack of studies evaluating apps and the great need to carry out rigorous studies that evaluate the functioning and usefulness of apps for pill identification and eventually to detect SCMs. MedSnap was the only app where the sensitivity and specificity to detect authentic and counterfeit drugs was assessed [21]. Most mobile apps were developed as pill identifiers, and yet developers state that the apps have the capacity to also detect SCMs (as an example of this would be the apps Medsnap and CheckFake) [22,23]. In real life these apps are mostly used to identify pills by elderly people or people with difficulties to recognize medications but not to identify SCMs. Although it cannot be considered a proxy of accuracy, we rated and compared the quality of the apps globally and by different domains. The global app quality mean MARS score from the 25 apps available on platforms was classified as acceptable (3.17) and it is similar to some review and patients experience studies for some mobile apps for others objectives [13,24–28]. Despite the aim of these apps, a low average score in the 'information dimension' (quantity and quality) was observed based on MARS. This is due partly because some of these apps do not seem to use a reliable or verifiable information source for medicines' attributes. Nevertheless, for the evidence-based question inside the 'information dimension', we found only two low-quality studies which evaluated the performance of four highly recognized apps (Epocrates, IBM Micromedex Drug Reference, Medscape and MedSnap) [20,21]. Regarding the MARS questions #15 and #16, which assessed the accuracy and comprehensiveness of the content and information, only 6 of 25 apps had a MARS score higher than 4. This could be a

concerning issue since 76% of the surveyed apps were not adequately backed up by any reliable information. As Kim et al, we also noted associations between the average user rating and the information dimension; this means that users may critically evaluate applications and these ratings can potentially be important tools for selecting apps [25].

Most of the identified mobile apps were aimed at consumers or general population, and just one was designed to be used by regulatory authorities, such as customs security personnel. Eight of the 25 apps (32%) were paid apps, and all the websites were open access. This proportion of paid apps was consistent with other studies that systematically reviewed the Google Play Store and Apple's App Store [13,24,25]. Eight of the 24 apps available in the stores had no average score from the users, but had a very low MARS mean score in our study (from 1.6 to 3.7). Like Kim et al, we found a correlation between the worst scores from the public and low scores in our MARS aesthetics and engagement dimensions [25].

SCMs are a worldwide problem with direct consequences for human health. The problem is much more severe in low- and middle-income countries due to their relatively weak health systems and regulatory processes. In an analysis of 215 misoprostol samples, 55% were within specifications, 85 (40%) were below average in 90% of the labelled content [29]. Of the 85 samples, 14 contained no misoprostol at all. There is a tangible threat to global health security, for example, in antiparasitic drugs (chiefly antimalarials) and antibiotics, increasing transmission, morbidity, mortality, and resistance. This is also the case for medications affecting maternal, neonatal and child health, like misoprostol or mifepristone. In this regard, mobile apps could be a valuable and inexpensive tool for patient empowerment and they have the potential to detect these drugs and save lives at a low cost. However, the fight against falsified and substandard drugs needs a multi-pronged approach with all the stakeholders, and although technology alone cannot solve the problem, it can be an important tool for consumers [11,14]. Those countries with lower regulatory capacity are the most vulnerable to SCM medications. The goal is to include guarantee of good manufacturing practices, and to impose audit control measures on manufacturing companies and distribution of medications [30,31].

While MARS score is helpful to indicate user-friendliness, there is a significant gap in rigorous evaluation of these apps. The most important limitation is that we cannot extrapolate the identification of a pill by a device to the categorization of a product as counterfeit or substandard. To find out whether the product it really of poor quality, additional confirmatory tests will be necessary. Additionally, we do not know how consumers use these applications or what their performance looks like in real life, so this makes conclusions about their accuracy in the detection of substandard-quality drugs difficult. Moreover, the reproducibility of the search on app platforms was not standardized, but iteration using keywords and related apps reduce the probability of missing apps. Finally, mobile apps have frequent updates, and new apps are published on a daily basis.

Our systematic review highlighted an important evidence gap in diagnostic accuracy of mobile apps detecting SCMs, and there is a need for primary studies addressing this issue. A unified global effort to address the important problem of counterfeit and substandard drugs is necessary. Our findings suggest that, although there is no single technology that can meet all the desired requirements for detecting SCMs, mobile apps could constitute a potential valuable real-world tool available to large number of potential users to counter the serious consequences of the SCM problem and help achieve the United Nations' Sustainable Development Goals.

## Supporting information

**S1 File. PRISMA checklist.**
(DOCX)

**S2 File. Search strategy.**
(DOCX)

**S3 File. NIH study quality assessment tools for observational studies.**
(XLSX)

## Acknowledgments

We would like to express our sincere thanks to our librarian Daniel Comandé, for leading the search strategy. In addition, we would like to thank Antonella Francheska Lavelanet for providing feedback and edits to this paper.

## Author Contributions

**Conceptualization:** Agustín Ciapponi, A. Metin Gülmezoglu, Ariel Bardach.

**Data curation:** Manuel Donato, Tomás Alconada.

**Formal analysis:** Agustín Ciapponi, Manuel Donato, Tomás Alconada, Ariel Bardach.

**Investigation:** Agustín Ciapponi, Manuel Donato, Tomás Alconada, Ariel Bardach.

**Methodology:** Agustín Ciapponi, Manuel Donato, A. Metin Gülmezoglu, Tomás Alconada, Ariel Bardach.

**Project administration:** Agustín Ciapponi, Ariel Bardach.

**Resources:** A. Metin Gülmezoglu.

**Supervision:** Agustín Ciapponi, A. Metin Gülmezoglu, Ariel Bardach.

**Validation:** Agustín Ciapponi, A. Metin Gülmezoglu.

**Writing – original draft:** Agustín Ciapponi, Manuel Donato, Tomás Alconada, Ariel Bardach.

**Writing – review & editing:** Agustín Ciapponi, A. Metin Gülmezoglu, Tomás Alconada, Ariel Bardach.

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
