## [Decision Letter · Decision Letter 0]

27 Oct 2020

PONE-D-20-22666

Mobile apps for Detecting Falsified and Substandard Drugs: A systematic review

PLOS ONE

Dear Dr. Ciapponi,

Thank you for submitting your manuscript to PLOS ONE. After careful consideration, we feel that it has merit but does not fully meet PLOS ONE’s publication criteria as it currently stands. Therefore, we invite you to submit a revised version of the manuscript that addresses the points raised during the review process.

We look forward to receiving your revised manuscript.

Kind regards,

Vijayaprakash Suppiah, PhD

Academic Editor

PLOS ONE

Journal Requirements:

2. Please clarify the last date of your search for articles and applications.

3. You have provided a table of characteristics of the mobile applications; however, you have not summarized the characteristics (and quality) of the two studies you included.

4. Neither publication bias nor study heterogeneity have been assessed. Please clarify why this is so.

5. In your PRISMA checklist, you indicate that you did not present results of any risk assessment of bias across studies. However, in the Methods section, you indicate that risk of bias was assessed. Please present a summary of your findings if any.

Reviewers' comments:

Reviewer's Responses to Questions

**Comments to the Author**

1. Is the manuscript technically sound, and do the data support the conclusions?

Reviewer #1: Partly

2. Has the statistical analysis been performed appropriately and rigorously? 

Reviewer #1: N/A

3. Have the authors made all data underlying the findings in their manuscript fully available?

Reviewer #1: Yes

4. Is the manuscript presented in an intelligible fashion and written in standard English?

Reviewer #1: Yes

5. Review Comments to the Author

Reviewer #1: My main concern with this paper is that it is not a systematic review of mobile apps for detecting falsified and substandard drugs (SF drugs). Rather, it is a review of 24 apps and 16 websites whose function is pill identification, and one app (MedSnap) whose function is to detect SF drugs. The authors used the MARS assessment tool to evaluate the usability of each app and website, but they didn’t actually assess any SF products. The authors point out that “we cannot extrapolate the identification of a pill by a device to the categorization of a product as counterfeit or substandard.” So, this manuscript reviewed the ease of use of “pill identification apps,” and that’s how it should be titled and pitched. I think the gaps in SF drug detection are serious and speak to a real problem in the world, but it’s not fair to evaluate apps designed for pill identification on this basis.

→change the title to remove the idea that the mss reviews mobile apps and websites whose function is to detect SF drugs

→add discussion of the pill identification task and its context

→add discussion of the technical challenges for detecting SF drugs via pill image analysis (this is why none of the apps/sites that rely on the user inputting pill color/shape/imprint data are capable of SF detection)

Most of the pill identification apps and websites, such as the highly rated Medscape website, ask the user to input pill imprint numbers, color, and shape, and match that information to a market-specific database. Only the most carelessly made fake products would fail to match at this level. The authors say (line 197) that “developers claim that the apps have the capacity to also detect SCMs”.

→I would like this statement to be backed up by data about which pill identifier products make this claim.

Tools such as MedSnap utilize pill photographs which can be compared to a library of known “good” pills via image analysis tools. This method detects subtle differences between the imprint or tablet size/shape/color of a genuine manufacturer and a counterfeit version. (That was the study they did in Laos on fake artesunate and coartem.)

→There is no evidence that this kind of image analysis can detect substandard products.

→The authors might consider discussing the CD3/CDx device, which is another image analysis tool that uses multispectral imaging of pharmaceuticals—there is a published field testing record for a range of SF products.

Ranieri N et al (2014) Evaluation of a new handheld instrument for the detection of counterfeit artesunate by visual fluorescence comparison. Am J Trop Med Hyg 91:920–924. https://doi.org/10.4269/ajtmh.13-0644

6. PLOS authors have the option to publish the peer review history of their article (what does this mean?). If published, this will include your full peer review and any attached files.

Reviewer #1: **Yes: **Marya Lieberman

---

## [Author Response · Author response to Decision Letter 0]

19 Nov 2020

Done

2. Please clarify the last date of your search for articles and applications.

Done: January 2020

 3. You have provided a table of characteristics of the mobile applications; however, you have not summarized the characteristics (and quality) of the two studies you included.

We found only two studies which evaluated the performance of four highly recognized apps (Epocrates, IBM Micromedex Drug Reference, Medscape and MedSnap).[20,21] All these apps (the focus of our systematic review) are described in Table 1. 

However, we expanded the narrative description of these studies and its quality as it can see below: 

“Our search found a cross-sectional study that intended to compare Epocrates, Medscape, IBM Micromedex and Google for clinical management information designed for healthcare professionals for a one-week period.[20] Medical students at an academic hospital in the United States used a score for satisfaction of search and user interface based on a 1 to 5 scale, with 1 representing lowest quality and 5 describing the highest quality. The study concluded that Medscape (Satisfaction 4.92) was the most preferred free mobile app evaluated due to its interactive and educational features, followed by IBM Micromedex and Epocrates (Satisfaction 4.58 and 4.42, respectively). This study was used to complete MARS question #19 (evidence base) about whether the app had been trialed. Further, we found a preprint study whit the objective to evaluate the sensibility and specificity of MedSnap, that is not available on the app stores. In the study,[21]”

“We used the NIH Study Quality Assessment Tools for the 2 included studies and in both cases a poor score was obtained, which means a high risk of bias. This demonstrates the lack of studies evaluating apps and the great need to carry out rigorous studies that evaluate the functioning and usefulness of apps for pill identification and eventually to detect SCMs. Likewise, the risk of bias assessment for RCTs could not be carried out since no study with this type of design was found.”

4. Neither publication bias nor study heterogeneity have been assessed. Please clarify why this is so.

We performed a tabular and narrative synthesis of each identified mobile app. Since a meta-analysis is not applicable and there is no positive or negative result of apps’ descriptive reports, publication bias couldn’t be assessed. However, our exhaustive search and inclusion criteria prevent omissions of relevant studies.

We considered that a formal assessment of heterogeneity was not applicable. However, we describe and discuss the differences found among included apps.

5. In your PRISMA checklist, you indicate that you did not present results of any risk assessment of bias across studies. However, in the Methods section, you indicate that risk of bias was assessed. Please present a summary of your findings if any.

We summarized in Table 1 and 2 and Figure 2 the general characteristics of the reviewed apps and the app quality mean MARS scores available on platforms.

The MARS score is the instrument to assess the apps quality.

We added the quality assessment of the only two studies using NIH instrument for these cross-sectional studies.

“We used the NIH Study Quality Assessment Tools for the 2 included studies and in both cases a poor score was obtained, which means a high risk of bias. This demonstrates the lack of studies evaluating apps and the great need to carry out rigorous studies that evaluate the functioning and usefulness of apps for pill identification and eventually to detect SCMs. Likewise, the risk of bias assessment for RCTs could not be carried out since no study with this type of design was found.”

Done

Reviewers' comments:

Reviewer's Responses to Questions

Comments to the Author

1. Is the manuscript technically sound, and do the data support the conclusions?

Reviewer #1: Partly

2. Has the statistical analysis been performed appropriately and rigorously?

Reviewer #1: N/A

3. Have the authors made all data underlying the findings in their manuscript fully available?

Reviewer #1: Yes

4. Is the manuscript presented in an intelligible fashion and written in standard English?

Reviewer #1: Yes

5. Review Comments to the Author

Reviewer #1: My main concern with this paper is that it is not a systematic review of mobile apps for detecting falsified and substandard drugs (SF drugs). Rather, it is a review of 24 apps and 16 websites whose function is pill identification, and one app (MedSnap) whose function is to detect SF drugs. 

Considering the fulfillment of different definitions of systematic reviews, we considered that our study is indeed a systematic review: 

*DARE: ≥ 4 criteria out of the first 5 (1-3 are mandatory)

#Oxman and Guyatt 1991: 1-4 & 6-7

#Cochrane Collaboration, CRD, MOOSE, Potsdam Consultation, QUOROM, AHRQ: 1-4 & 6-8

1. Were inclusion/exclusion criteria reported? Yes

2. Was the search adequate? Yes

3. Were the included studies synthesised? Yes, a Tabular and narrative synthesis

4. Was the validity of the included studies assessed? 

Yes, we used the NIH for the only 2 studies identified. Additionally, the other were descriptions of apps that were classified by MARS.

5. Are sufficient details about the individual included studies presented? Yes, in tables and figures

6. Was the data extraction process adequate? Yes

Data extraction and risk of bias (quality) assessment were also performed independently by these reviewers, with oversight from two senior reviewers (AC and AB).

7. Was the study selection process adequate? Yes

All unique articles were independently assessed by two reviewers (MD and TA) based on title and abstract. Those marked for inclusion, or whose title and abstract were not sufficient to determine inclusion, were then reviewed using the full text.

8. Was ‘PICO’ used to focus the question(s)?

Not completely applicable, but PICO relevant components are derived from the presented objectives

*DARE. (Accessed 04/08/2011, 2011, at http://www.crd.york.ac.uk/cms2web/AboutDare.asp.)

# Sander L, Kitcher H. Systematic and Other Reviews: Terms and Definitions Used by UK Organizations and Selected Databases. Systematic Review and Del-phi Survey. In: National Institute for Health and Clinical Excellence. London; 2006.

The authors used the MARS assessment tool to evaluate the usability of each app and website, but they didn’t actually assess any SF products. The authors point out that “we cannot extrapolate the identification of a pill by a device to the categorization of a product as counterfeit or substandard.” So, this manuscript reviewed the ease of use of “pill identification apps,” and that’s how it should be titled and pitched. I think the gaps in SF drug detection are serious and speak to a real problem in the world, but it’s not fair to evaluate apps designed for pill identification on this basis.

→change the title to remove the idea that the mss reviews mobile apps and websites whose function is to detect SF drugs

Thank for such appropriate comment. We searched for direct assessments of SF products using these apps and this define the title and aims of our study. Unfortunately, we did not find any reported experiments of real-world SFs using these technologies. We highlighted this evidence gap and we recommended caution to interpret the pill identification quality scores due to the indirectness of the findings. A sentence was added in the result section to clarify this point.

‘’These reports referred to the usability and the ability of apps to correctly identify pills. However, did not find direct assessment of SF products in the real-world.”

→add discussion of the pill identification task and its context

We added this sentence in the discussion section

“In real life these apps are mostly used to identify pills by elderly people or people with difficulties to recognize medications but not to identify SCMs.”

→add discussion of the technical challenges for detecting SF drugs via pill image analysis (this is why none of the apps/sites that rely on the user inputting pill color/shape/imprint data are capable of SF detection)

While detecting SF drugs via pill image analysis does not allow proper SC drugs detection it might help as an initial screening test, particularly with negative results.

Most of the pill identification apps and websites, such as the highly rated Medscape website, ask the user to input pill imprint numbers, color, and shape, and match that information to a market-specific database. Only the most carelessly made fake products would fail to match at this level. The authors say (line 197) that “developers claim that the apps have the capacity to also detect SCMs”.

→I would like this statement to be backed up by data about which pill identifier products make this claim.

Two apps claim to have the capacity to detect SCMs: Medsnap and CheckFake

Tools such as MedSnap utilize pill photographs which can be compared to a library of known “good” pills via image analysis tools. This method detects subtle differences between the imprint or tablet size/shape/color of a genuine manufacturer and a counterfeit version. (That was the study they did in Laos on fake artesunate and coartem.)

→There is no evidence that this kind of image analysis can detect substandard products.

MedSnap detects SF drugs via pill image analysis. Although it does not allow a proper SC drugs detection, it still might help as an initial screening test, that could be particularly useful if it discard SC drugs.

→The authors might consider discussing the CD3/CDx device, which is another image analysis tool that uses multispectral imaging of pharmaceuticals—there is a published field testing record for a range of SF products.

Ranieri N et al (2014) Evaluation of a new handheld instrument for the detection of counterfeit artesunate by visual fluorescence comparison. Am J Trop Med Hyg 91:920–924. https://doi.org/10.4269/ajtmh.13-0644

Thank you but our systematic review was focused only on mobile apps.

---

## [Decision Letter · Decision Letter 1]

17 Dec 2020

PONE-D-20-22666R1

Mobile apps for detecting falsified and substandard drugs: A systematic review

PLOS ONE

Dear Dr. Ciapponi,

Thank you for submitting your manuscript to PLOS ONE. After careful consideration, we feel that it has merit but does not fully meet PLOS ONE’s publication criteria as it currently stands. Therefore, we invite you to submit a revised version of the manuscript that addresses the points raised during the review process.

We look forward to receiving your revised manuscript.

Kind regards,

Vijayaprakash Suppiah, PhD

Academic Editor

PLOS ONE

Reviewers' comments:

Reviewer's Responses to Questions

**Comments to the Author**

1. If the authors have adequately addressed your comments raised in a previous round of review and you feel that this manuscript is now acceptable for publication, you may indicate that here to bypass the “Comments to the Author” section, enter your conflict of interest statement in the “Confidential to Editor” section, and submit your "Accept" recommendation.

Reviewer #1: (No Response)

2. Is the manuscript technically sound, and do the data support the conclusions?

Reviewer #1: Yes

3. Has the statistical analysis been performed appropriately and rigorously? 

Reviewer #1: N/A

4. Have the authors made all data underlying the findings in their manuscript fully available?

Reviewer #1: Yes

5. Is the manuscript presented in an intelligible fashion and written in standard English?

Reviewer #1: Yes

6. Review Comments to the Author

Reviewer #1: PONE-D-20-22666R1 Mobile apps for detecting falsified and substandard drugs: A systematic review

I am flabbergasted by the general lack of documentation and regulatory oversight of these products and think it’s worthwhile for the mss to be published to highlight this evidence gap.

I’m still having a problem with the title. The authors conducted a systematic review of apps and websites that are used for pill identification. Only two of them make any claims to detect falsified or substandard products, and only one of those has any evidence base to support the claims. Can you have a systematic review of one manuscript?

Maybe, call it “a systematic review of pill identification apps with potential utility for detecting SF drugs”?

Please add the references to the MedSnap and CheckFake distributor’s claims of SF detection to the mss.

From the app store entry, CheckFake is a bar code checker, not a pill checker, and does not seem to be supported by or connected with any company—I didn't see a website or published literature on it. Pill checkers need to be maintained and upgraded as new brands or drugs enter the market, or as the app is used in markets that offer different brands. In the future it would be interesting to evaluate whether each app is "really" available in different regions, eg, whether there is a company behind it to support users and upgrade the app and whether the product really works for the brands found in Europe, America, Africa, etc.

7. PLOS authors have the option to publish the peer review history of their article (what does this mean?). If published, this will include your full peer review and any attached files.

Reviewer #1: **Yes: **Marya Lieberman

---

## [Author Response · Author response to Decision Letter 1]

18 Dec 2020

Reviewer #1: PONE-D-20-22666R1 Mobile apps for detecting falsified and substandard drugs: A systematic review

Point by point answers

I am flabbergasted by the general lack of documentation and regulatory oversight of these products and think it’s worthwhile for the mss to be published to highlight this evidence gap.

Thank you for the comment, we have reflected now this key message in the conclusions of the abstract 

“In conclusion, we found a remarkable evidence gap about the accuracy of mobile apps in detecting SCMs.”

The conclusions of the manuscript already clearly reflected this concept

“Our systematic review highlighted an important evidence gap in diagnostic accuracy of mobile apps detecting SCMs, and there is a need for primary studies addressing this issue.”

I’m still having a problem with the title. The authors conducted a systematic review of apps and websites that are used for pill identification. Only two of them make any claims to detect falsified or substandard products, and only one of those has any evidence base to support the claims. Can you have a systematic review of one manuscript?Maybe, call it “a systematic review of pill identification apps with potential utility for detecting SF drugs”?

Dear reviewer, our research question was about mobile apps for detecting falsified and substandard drugs and our protocol was registered in PROSPERO (CRD42020163075) under this name. We consider a good research practice to be consistent with the protocol, regardless of the findings finally obtained.

The name of a systematic review refers to the research question that guide the search strategy of a systematic review. Is not only possible that a systematic review has only one included study, but may be a dessert systematic review with cero included studies (There are multiple examples in the Cochrane Library). Of course, as you suggested, we have reinforced the first key message highlighting the identified evidence gap. 

Additionally, the pill identification functionality could indirectly help to detect falsified and substandard drugs.

This limitation in the identified evidence is also clearly discussed in the manuscript.

Please add the references to the MedSnap and CheckFake distributor’s claims of SF detection to the mss.

Done. These are the references included in the manuscript now:

22. Medsnap Verify Services. 2020 [cited 17 Dec 2020]. Available: https://www.medsnap.com/verify/

23. Google play. CheckFake (dlt.sg App for drug counterfeit). 2020 [cited 17 Dec 2020]. Available: https://play.google.com/store/apps/details?id=sg.dlt.checkfake&hl=es_419

From the app store entry, CheckFake is a bar code checker, not a pill checker, and does not seem to be supported by or connected with any company—I didn't see a website or published literature on it. Pill checkers need to be maintained and upgraded as new brands or drugs enter the market, or as the app is used in markets that offer different brands. In the future it would be interesting to evaluate whether each app is "really" available in different regions, eg, whether there is a company behind it to support users and upgrade the app and whether the product really works for the brands found in Europe, America, Africa, etc.

Thank you for the comment. We have corrected the information about the CheckFake app. We had to include it because this reference met our inclusion criteria, however our assessment of the quality and reliability of the information of this app was evaluated with the worst of the values of the MARS questionnaire.

---

## [Decision Letter · Decision Letter 2]

13 Jan 2021

Mobile apps for detecting falsified and substandard drugs: A systematic review

PONE-D-20-22666R2

Dear Dr. Ciapponi,

We’re pleased to inform you that your manuscript has been judged scientifically suitable for publication and will be formally accepted for publication once it meets all outstanding technical requirements.

Kind regards,

Vijayaprakash Suppiah, PhD

Academic Editor

PLOS ONE

Reviewers' comments:

Reviewer's Responses to Questions

**Comments to the Author**

1. If the authors have adequately addressed your comments raised in a previous round of review and you feel that this manuscript is now acceptable for publication, you may indicate that here to bypass the “Comments to the Author” section, enter your conflict of interest statement in the “Confidential to Editor” section, and submit your "Accept" recommendation.

Reviewer #1: All comments have been addressed

2. Is the manuscript technically sound, and do the data support the conclusions?

Reviewer #1: (No Response)

3. Has the statistical analysis been performed appropriately and rigorously? 

Reviewer #1: (No Response)

4. Have the authors made all data underlying the findings in their manuscript fully available?

Reviewer #1: (No Response)

5. Is the manuscript presented in an intelligible fashion and written in standard English?

Reviewer #1: (No Response)

6. Review Comments to the Author

Reviewer #1: (No Response)

7. PLOS authors have the option to publish the peer review history of their article (what does this mean?). If published, this will include your full peer review and any attached files.

Reviewer #1: **Yes: **Marya Lieberman https://orcid.org/0000-0003-3968-8044

---

## [Editor Report · Acceptance letter]

26 Jan 2021

PONE-D-20-22666R2 

Mobile apps for detecting falsified and substandard drugs: A systematic review 

Dear Dr. Ciapponi:

I'm pleased to inform you that your manuscript has been deemed suitable for publication in PLOS ONE. Congratulations! Your manuscript is now with our production department. 

Kind regards, 

on behalf of

Dr. Vijayaprakash Suppiah 

Academic Editor

PLOS ONE